



# Progress of the RADIX fast access drilling system

Jakob Schwander, Thomas F. Stocker, Remo Walther, Samuel Marending

Climate and Environmental Physics, Physics Institute, University of Bern, Sidlerstrasse 5, 3012 Bern & Oeschger Centre for Climate Change Research, University of Bern, Switzerland

*Correspondence to*: Thomas F. Stocker (thomas.stocker@unibe.ch)

**Abstract.** A rapid access drilling system has been designed, realized and tested on polar ice sheets. A 20-mm access hole is drilled with a hydraulic coiled tubing system where the tube consists of a plastic hose. The ice cuttings are flushed to the surface in forward circulation, i.e. between hose and borehole wall, and can be sampled for analysis. A dedicated logger with 15 mm diameter is deployed into the hole for measuring the geometric orientation of the hole, the temperature profile and the

dust content in the surrounding ice (optical dust logger). The equipment has been tested and improved during 6 field projects in Greenland and Antarctica. It is designed to ultimately drill quickly through a 3000 m ice sheet. Routine performance has been established to a depth of 300 m.

## 1. Introduction

In recent decades, ice core research has created an extensive knowledge base for climate history and glaciology. Yet in polar

ice sheets there is the potential to extend that knowledge for example back in time (oldest ice projects) or with higher resolution for special time ranges or geographic areas (ice streams, WAIS etc). Deep drilling projects are costly and time consuming. Fast access ice drilling could be used to find suitable drilling sites in advance of new deep drilling projects and therefore reduce costs and avoid drill sites with unfavorable conditions. Several fast access drilling projects have been proposed and tested (Alemany et al., 2014; Clow and Koci, 2002; Goodge et al., 2021; Rix et al., 2019)

The University Bern RADIX (Rapid Access Drilling and Ice eXtraction) project has been designed for drilling a very small diameter hole with a depth range of up to 3 km. It aims at minimal overall resources and weight. The concept of RADIX and first feasibility tests have been published by Schwander et al. (2014). The present version of RADIX is still very close to this published concept, but many details have been improved or added based on a series of previous test drillings in Greenland, Antarctica and in the laboratory. In the 2021/2022 Antarctic season we have drilled the so far deepest hole of 320 m at Little

Dome C. Here we present the current status of the RADIX system and share our experiences and results from the field work and from laboratory tests that were essential for the technical development of RADIX. Table 1 summarizes key parameters of the RADIX system.




| Parameter | Value |
|---|---|
| Total weight without fluid (26 colis) | 2150 kg |
| Total volume (packed) without fluid | 8 m³ |
| Firn drill | |
| - hole diameter | 40 mm |
| - Motor voltage | 48 VDC |
| - Max. current | 1.9 A |
| - Average drill current | 0.3 A |
| - Nominal rotation speed | 4.76 rps |
| - Nominal drilling speed | 15 m/h |
| - Maximal depth | 120 m |
| - PA vacuum hose o.d. / i.d. | 22 mm/ 19 mm |
| - Vacuum pump flow rate | 60 m3/h |
| End pressure of vacuum pump | 100 mb |
| RADIX drill | |
| - Hole diameter | 20 mm |
| - Nominal rotation speed | 12 rps |
| - Pitch | 0.75 mm |
| - Nominal drilling speed | 9 mm/s |
| - Moineau motor type | 5/6 lobes |
| - Displaced volume/revolution | 2.96 cm³ |
| Maximum torque (@60 bar) | 1.6 Nm |
| Fluid system | |
| - Fluid type | Polysiloxane MD2M |
| - Density @ -55°C | 930 kg/m3 |
| - Viscosity @ 20°C | 1.5 cSt |
| - Viscosity @ -55°C | 10 cSt |
| - Min. fluid demand for 3000 m hole | 1100 L |
| - Max. pressure (Speck triplex pump) | 220 bar |
| - Flowrate (adjustable by VFD) | 1...6 L/min |
| - Hydraulic hose o.d./i.d. | 13.6 mm/8 mm |
| - Hydraulic hose length | 3100 m |
| - Hydraulic hose pressure rating | 295 bar |
| - Hydraulic hose burst pressure | 1480 bar |
| Hydraulic hose tensile strength | 6000 N |

**Table 1: Radix specifications**

## 2. RADIX components (2021 version)

### 2.1 Shelter

We use an inflatable 7m x 7m x 5m white single layer tent (Axion) as drill shelter (Fig. 1). The tent has an inflatable tube in each corner (legs) and an inflatable frame of 4 tubes for stabilizing the upper part of the tent. 4 side panels are zipped between the legs. The shelter is set up in a few hours, including securing for snow and wind. Two of the 4 side panels are





equipped with zipper doors for access and temperature control. At average daily temperatures of -35°C in Antarctica the tent
provided a calm interior working space with temperatures close to the freezing point. One door was usually left open to
prevent melting.

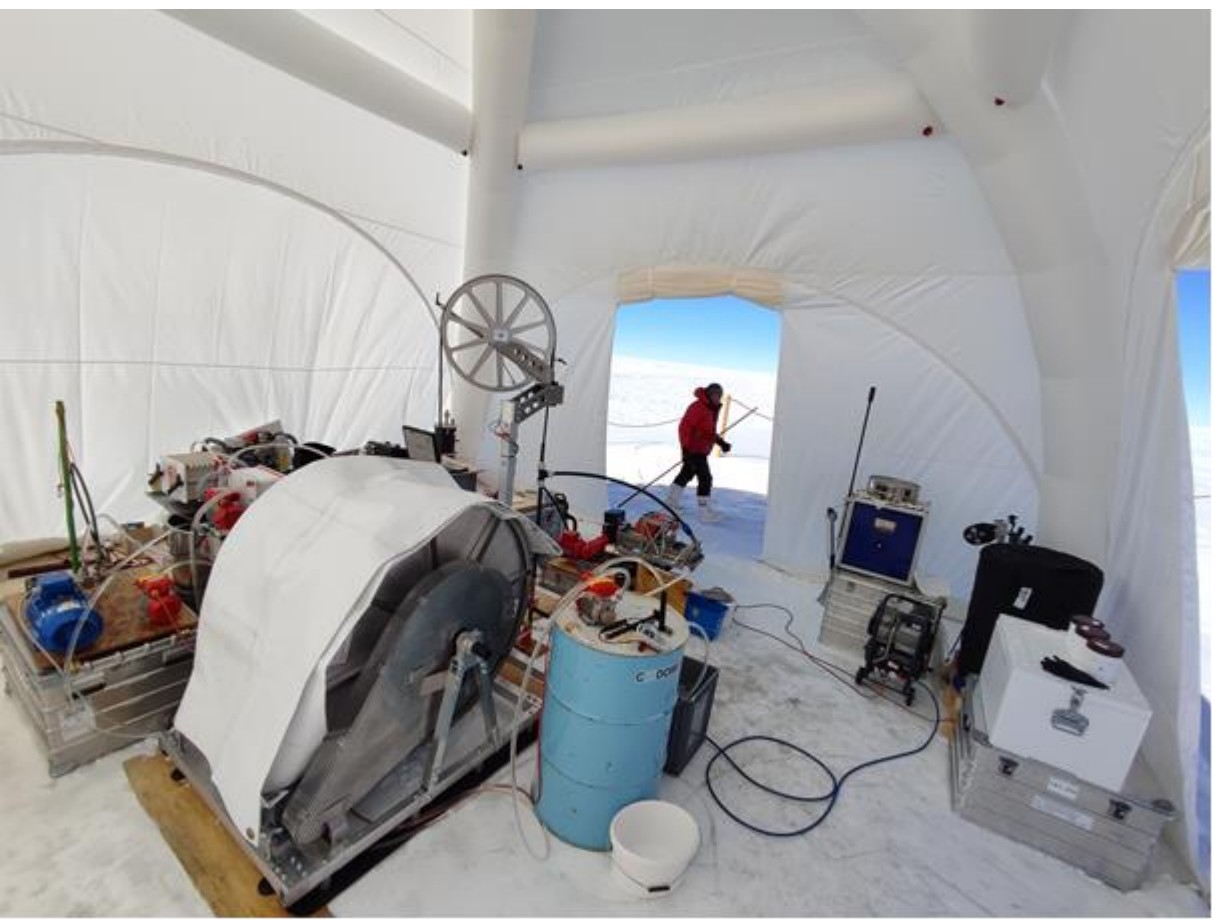

**Figure 1: The RADIX system is housed in an inflatable, wind-proof shelter (dimensions: 7m x 7m x 5m) from Axion (www.axion4event.com).**

**2.2 Firn drill**

In order to set casing and packer a 40-mm diameter hole is drilled from the surface to the ice below the bubble close-off
depth. The first 1.5 m are cut out with an aluminum pipe that is forced exactly vertically into the snow. Down to
approximately 6 m we melt the hole with a heated copper device, which has a shape similar to a plumb bob (Fig. 2).


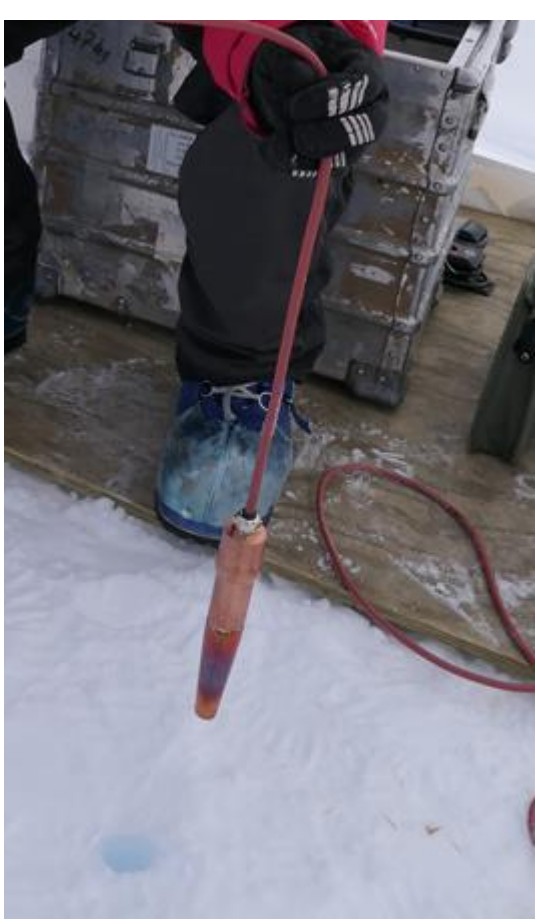

**Figure 2: Heated copper "plumb bob" for melting and glazing first 6 m of firm hole.**

The upper part of the bob has a spherical bead. During melting, the bead hangs in the firn, allowing the melter to pivot to the vertical position by gravity, keeping the melting direction very close to vertical. While melting, it also glaces the hole wall, thus making it mechanically more stable and reducing the risk of snow falling into the hole. The remaining hole is then drilled with a dedicated electro-mechanical drill (Fig. 3). Two 40 W motors (26 mm diameter) and a planetary gear driving the 40-mm drill bit are housed in a stainless steel pipe of 32 mm outer diameter connected to a 22 mm/19 mm (outer/inner diameter, henceforth), 120 m long vacuum pipe (polyamide tubing). The electrical power line is taped to the outside of the vacuum pipe. A 2 mm clearance between motors and the steel housing allows the ice cuttings to pass from the drill bit to the vacuum pipe. At the entrance to the drill housing, behind the bit, a coffee mill like rotary cutter ensures that only small ice cuttings can enter the vacuum line. A Zephyr claw vacuum pump C-VLR 60 (60 m3/h) provides enough flow for transporting the cuttings to the surface. A cyclonic separator between the vacuum line and pump collects the ice chips in a tank with a capacity for approximately 10 m drilling depth. The drill is lowered on the vacuum line and controlled by hand.



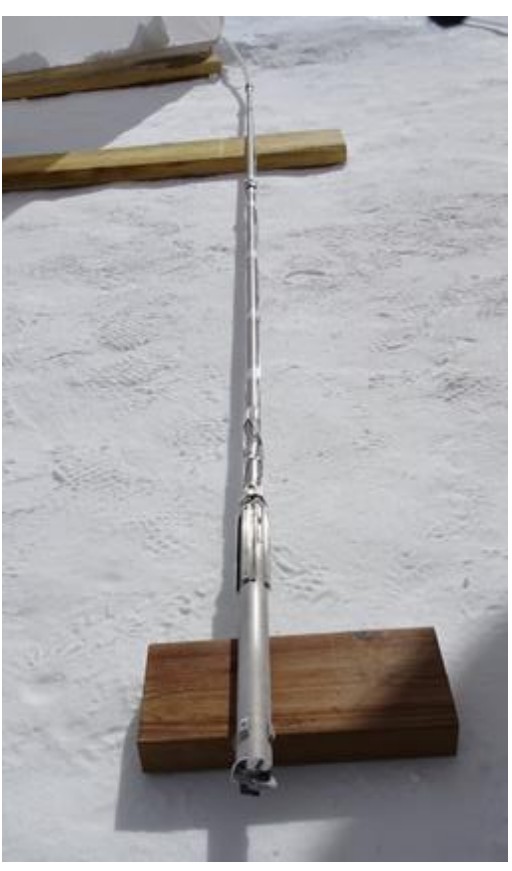

**Figure 3: 40-mm firn drill on vacuum line for lifting ice cuttings. All pieces are manufactured in the workshop of the Physics**
**Institute, University of Bern.**

**2.3 Firn hole casing and packer**

We use a 25 mm/21 mm HD-PE water pipe for casing the firn hole. As the total weight of 100 m casing is only about 14 kg it can be lowered or raised by hand. The casing is sealed to the solid ice at the bottom of the casing hole by means of a packer. The packer is made with a 0.7 m long rubber hose mounted on a stainless steel tube (Fig. 4). The casing pipe is fed

through the packer and screwed with thread sealant to an adapter mounted on the lower end of the steel tube. The rubber hose is inflated through a 3/2 mm pneumatic hose taped to the outside of the casing. The low inherent physical stability of the thin walled casing pipe causes the pipe to bend under its own weight. Therefore the pipe must be clamped to a lifting jack mounted on the drill tower and stretched in order to remove the bending in the hole. The packer is pressurized to approximately 3 bar above the hydrostatic pressure of the fluid at the level of the packer. The pressure is maintained by

compressed air from a 7 liter flask equipped with a pressure control valve. A manifold on the flask allows refilling during operation by means of a breathing air compressor. The casing extends about 1.3 m above the surface. During drilling the rising fluid exits through a side outlet tee at 1 m above the surface.



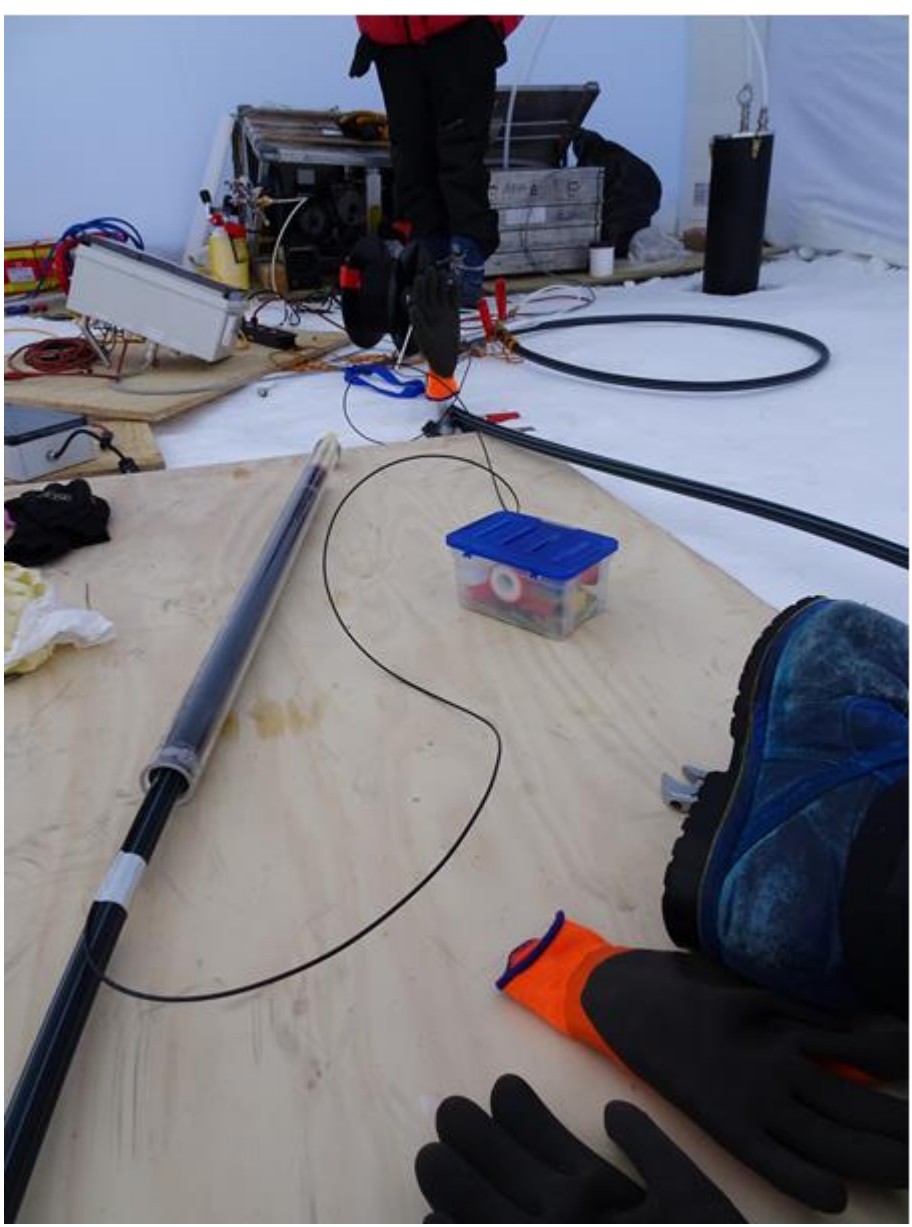

**Figure 4: Packer inserted into a Plexiglas tube to test inflation. The inner diameter of the tube is identical to the borehole**
**diameter.**

## 2.4 RADIX drill, anti-torque system and flow controller

Here we describe the RADIX drill components from the top to the bottom, in the direction of drilling. The drill (Fig. 5) is
connected to a hydraulic hose with a rotary union. Below this adapter the drill is centered and stabilized against rotation by
an anti-torque. The anti-torque is made with three vertical spring loaded sharp skates with a rake angle on the cutting edge of
-40 degrees. Below the anti-torque a pressure relief valve that can be set between 30 and 70 bar maintains a constant pressure




while drilling and insures fluid flow into the hole even in case of a blocked hydraulic motor. This is important to clear the hole from ice chips.

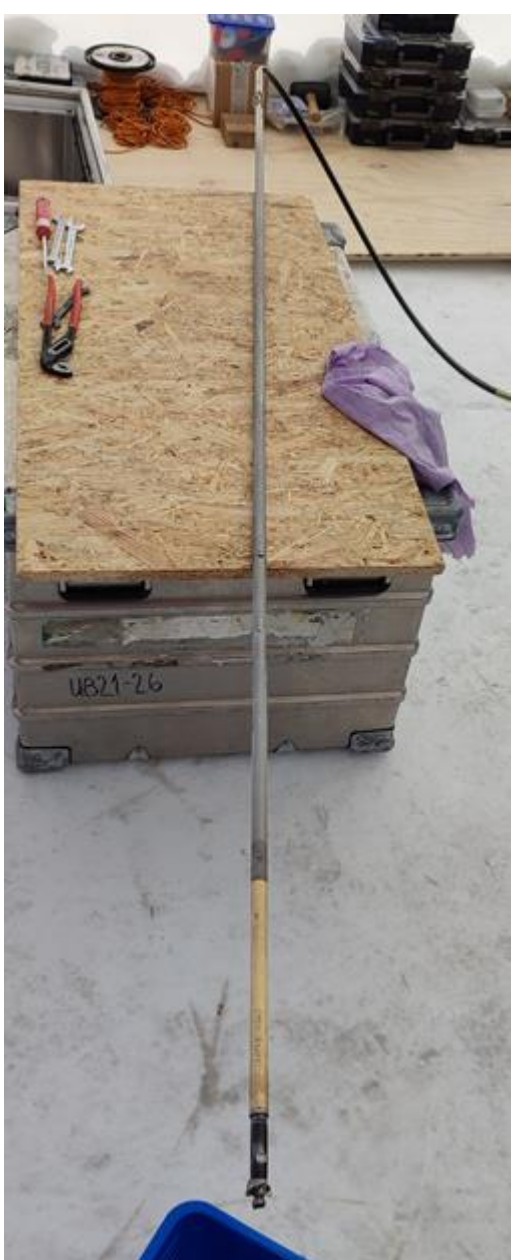

**Figure 5: RADIX fluid drill with tungsten extension tubes, connected to the hydraulic hose (top)**


The next component is a mechanical flow controller that provides an even flow and therefore a controlled rotation speed of the hydraulic motor. The operation principle of the flow controller is a fluid jet deflected on a spring loaded pin (Fig. 6). The



momentum change caused by the deflection moves the pin toward the spring and controls the flow by closing side channels in the pin housing.

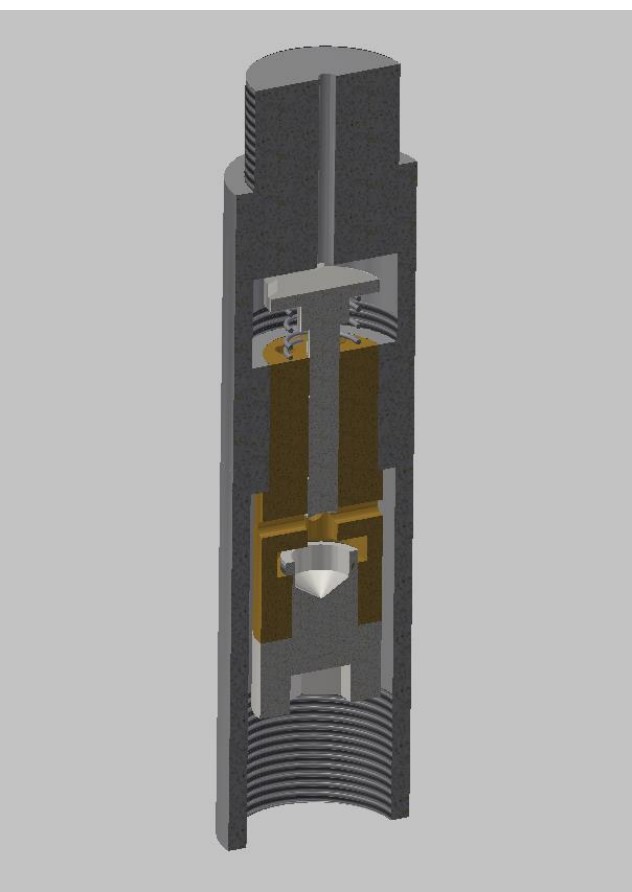


**Figure 6: Cross section through mechanical flow controller. The direction of flow is from top to bottom.**

Between the flow controller and the hydraulic motor we have inserted six sections of 15/5 mm tungsten tube with a total length of 1.58 m. These extensions add weight and reduce the tilt of the drill relative to the hole axis. Three centering knobs,

which extend exactly to the diameter of the hole are mounted on one of the tungsten sections. Besides reducing the overall bending of the drill they let the lower end of the drill hang downward. Having these knobs at a well determined position they provide an auto-correction to verticality for an inclined hole.

The hydraulic motor is a 5/6 lobes Moineau motor (mud motor) with a nominal rotation speed of 10 to15 rotations per second and a torque of about 1 Nm. The exact rotation speed depends on the drilling torque and the leak flow, which varies

with the fluid viscosity and thus with temperature. The setting of the pressure relief valve has also some influence on the flow rate and therefore on the rotation speed.



Both stator and rotor are made of maraging steel. Raw parts were laser sintered on a 3-D printer. The stator hole was then formed by spark erosion, the rotor was micro machined on a CNC milling machine, both with an accuracy of 1/100 mm. Rotor and stator are then polished against each other with a 800 grit boron carbide paste. After providing the connection
threads the parts are hardened and nitrated. Finally the parts are polished again against each other with a 3 μm diamond suspension. The eccentric rotation of the motor is transmitted to the bearing section via a flexible shaft.

The drill bit has a hollow shaft so that the drilling fluid exits at the very tip of the bit. The bit consists of a two stage drill (Fig. 7). The front center bit has a diameter of 10 mm and has two step cutting edges. On the outside, the center bit further cuts a thread with 0.75 mm pitch, which determines the penetration rate of the drill. The raw center bit has been 3-D printed
in maraging steel. The thread was then micro-machined on a CNC milling machine. The center bit is mounted into a 20 mm modified Forstner bit that widens the hole from 10 mm to 20 mm with two step cutting edges.

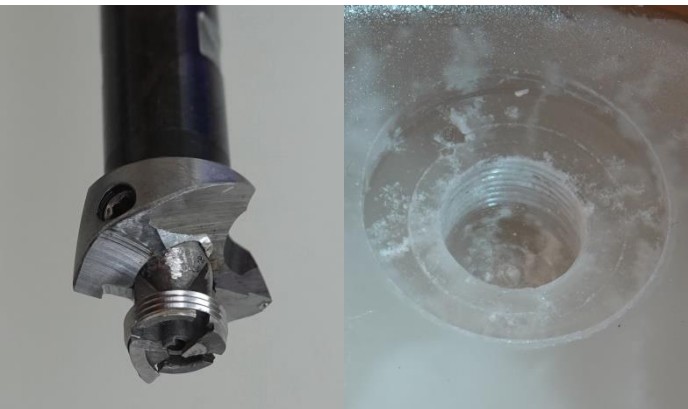

**Figure 7: RADIX 20-mm drill bit with 10-mm thread cutting center bit (left). Test drilling in paraffin wax (right).**

**2.5 Coiled tubing system**

The hydraulic hose of 13.6 mm/ 8 mm is 3100 m long (Aramid reinforced Hytrel 5556, from Kutting, UK). From the drill hole it passes over a 2-m circumference sheave mounted on a 2-m tower and is then coiled on a motor driven drum with rotary union for the inlet of the fluid. The sheave support is mounted on a bearing that allows horizontal swiveling about the axis of the casing pipe. This pivoting lets the hose to follow the coiling on the drum without the need of a special level wind
system. The feed of the hose is measured by a depth counter on the sheave. The specific weight of the hose is slightly higher than the one of the drilling fluid, providing a free fall sinking speed of about 0.15 m/s. The maximum coiling speed of the drum is about 0.6 m/s. A load pin in the bearing of the sheave measures the drag on the hose. The signal of the load pin can be looped to the winch drive and controls feed of the hose in automatic drilling mode. In this mode the drag on the hose is held at a preset value.



## 2.6 Fluid system and chip separation

We use 1.5 cSt Silicone Oil (MD2M-Polysiloxane). Its average in situ density matches the average in situ ice density at Dome Concordia very closely. Silicone oil from standard barrels is transferred by a level switch controlled pump to a 150 L drum (clean reservoir) (Figs. 8, 10). From there a booster pump feeds the fluid to the 220 bar high pressure triplex piston pump. A safety pressure relief valve at the outlet protects pump and downstream components. The high pressure pump is connected to the rotary union of the hose drum. After passing through the hydraulic hose and the drill the fluid carries the ice chips to the surface. The chips loaded fluid exits through the side outlet tee near the top of the casing and flows onto a slotted wire wedge screen (Fig. 9). The screen is mounted on an adjustable vibrator, which also effects the forward movement of the slurry into the screw press where the separated chips are further dried.

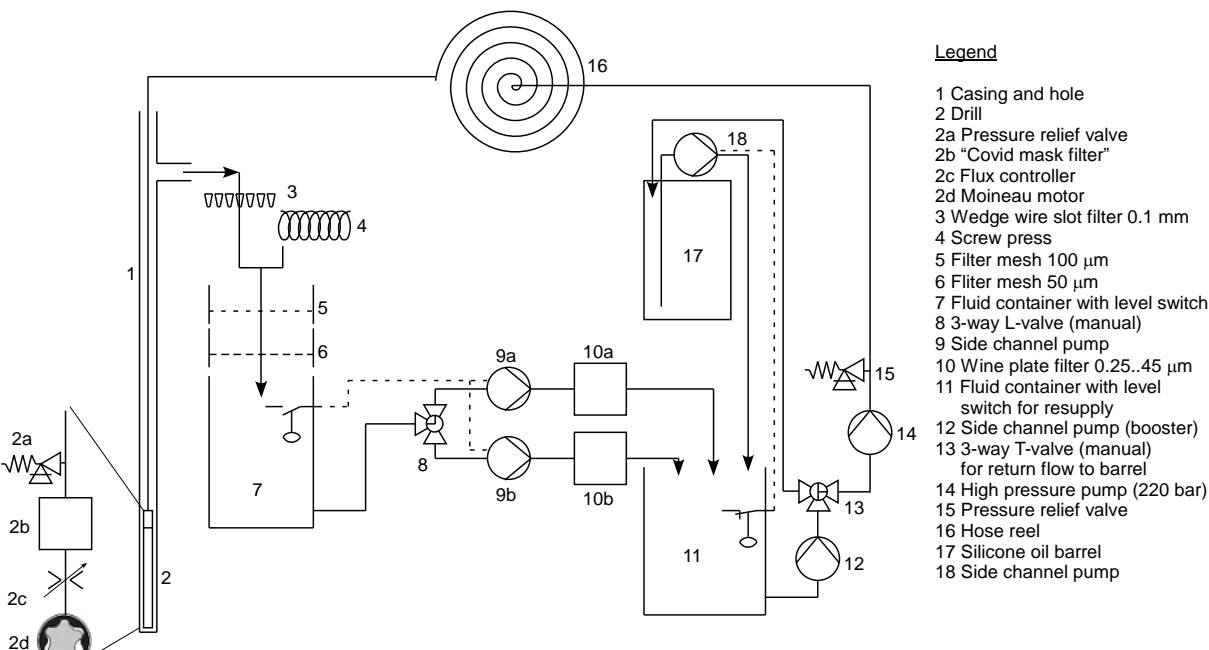

Legend

1 Casing and hole
2 Drill
2a Pressure relief valve
2b "Covid mask filter"
2c Flux controller
2d Moineau motor
3 Wedge wire slot filter 0.1 mm
4 Screw press
5 Filter mesh 100 μm
6 Fliter mesh 50 μm
7 Fluid container with level switch
8 3-way L-valve (manual)
9 Side channel pump
10 Wine plate filter 0.25..45 μm
11 Fluid container with level
    switch for resupply
12 Side channel pump (booster)
13 3-way T-valve (manual)
    for return flow to barrel
14 High pressure pump (220 bar)
15 Pressure relief valve
16 Hose reel
17 Silicone oil barrel
18 Side channel pump

**Figure 8: Schematics of RADIX fluid circulation**





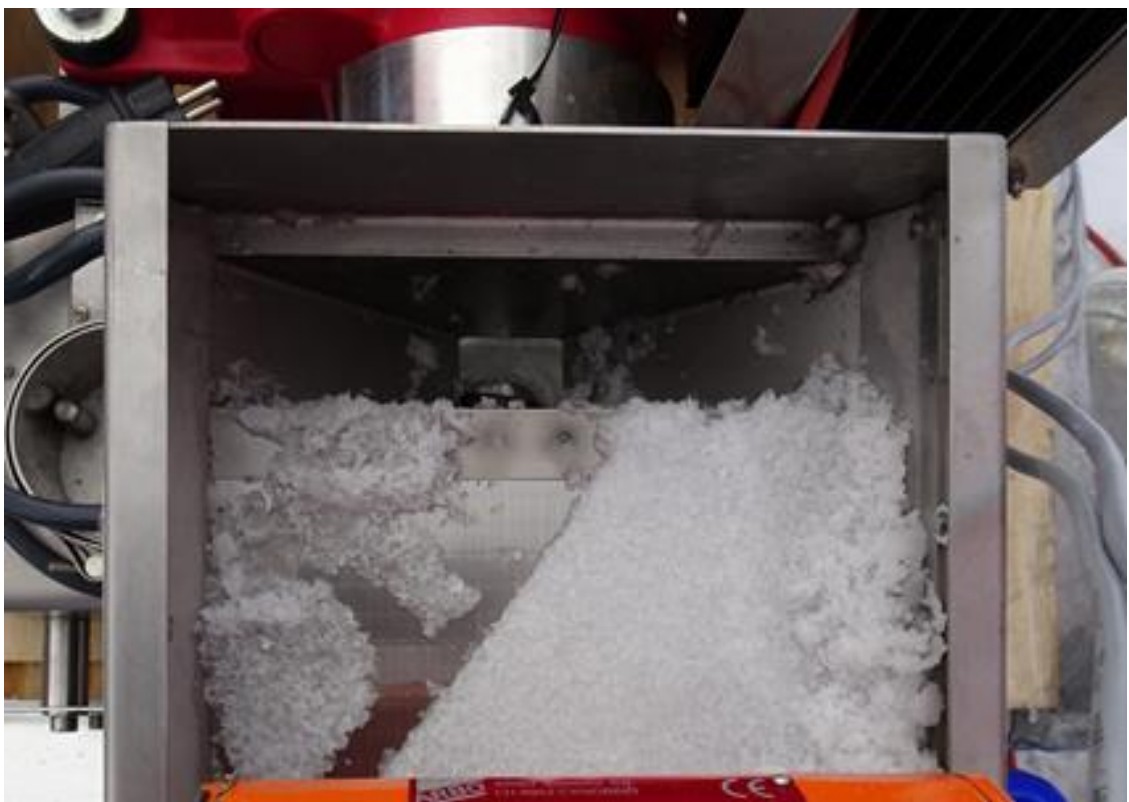

**Figure 9: Fluid-chips separation on slotted wire wedge screen. The slurry is moving upwards on the vibrating screen and falls into the screw press below.**


The fluid from the slotted screen and the screw press then passes a 100 μm and a 50 μm test sieve placed on a 50 L drum. The sieves are regularly swapped and cleaned. From the drum the fluid is transferred by a level switch controlled pump to a fine fluid filter (wine plate filter 0.25 to 45 μm) from where the clean fluid drips into the 150 L drum and thus closes the fluid cycle. The ice chips from the test sieves and the ice pellets from the screw press are melted in a 200 L heated drum

where water and silicone oil separates. The oil is transferred back to the fluid system after passing the test sieves and fine filter (Figs. 8, 10).

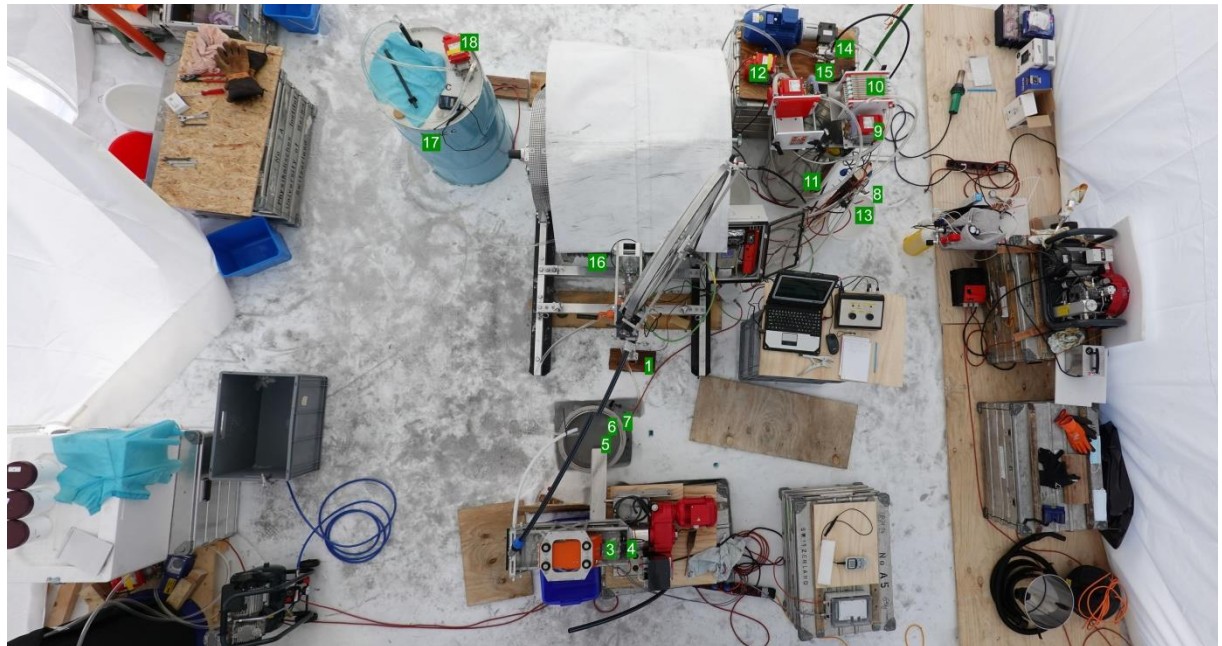

**Figure 10: Bird's eye view of the RADIX system. Component numbers correspond to legend in Fig. 8.**

### 2.7 Fluid recovery

A small rubber packer can be mounted at the end of the previously emptied hydraulic hose for removing fluid from the hole. The hose with packer is lowered into the fluid filled hole as deep as it sinks by its own weight. Then compressed air is blown through the hydraulic hose, which finally exits below the packer. A pressure relief valve in the packer maintains a slight overpressure in the rubber packer hose, which consequently creates a seal between packer and hole. When the compressed air reaches the hydrostatic pressure at the level of the packer, the hose is slowly lifted together with the fluid column above

the packer. The lifted fluid can then be collected through the tee outlet at the top of the casing. Several recovery cycles are needed to empty a deep hole.

### 2.8 Ice sampling

Instead of melting the ice pellets from the screw press they can be sampled from specific depths for further analysis. The pellets contain about 20 % of fluid.

### 2.9 Bore hole logger

The RADIX logger is essentially independent of the drilling. It is deployed into the hole with its own winch and cable (Fig. 11). However we use the same tower, sheave and depth counter as for drilling. A software adjustment is made in the depth counter to account for the smaller diameter of the cable. The logger is battery powered (8 AAA Lithium batteries). The logging data are transmitted in real time over a single mode optical fiber in a Kevlar reinforced HD-PE cable with 3.5 mm





outer diameter. The logging electronics is housed in a 15 mm/ 12 mm stainless steel tube. The high pressure seals are designed to withstand at least 300 bar.

The logger monitors temperature, 3-D acceleration, 3-D magnetic field, dust in the surrounding ice, system voltages and currents. Data are measured and transmitted at a rate of approximately 10 s$^{-1}$. They are stored in a database on a notebook together with the measured values of the depth counter. The data of the database are displayed in real time on the notebook

screen.

The principle of the dust logger is based on the optical dust logger developed by Bay et al. (2001). Here we transmit blue light with a 2.4 W LED (405 nm) into the ice and measure the light scattered back by the dust particles in the ice with a 10-mm photomultiplier tube (Hamamatsu R1635). The light is transmitted and received through dedicated sapphire windows and a mirror soldered into the pressure housing. Both windows point to the same side of the logger and are separated by 0.38

m.

The outside of the housing of the logger has been painted with a black coating to minimize light travelling from the LED to the photomultiplier tube in the gap between hole wall and logger case. The sensitivity of the photomultiplier reaches down to single photons. In order to cover the expected range of backscattered light the output power of the LED as well as the high voltage of the photomultiplier is cyclically switched between two values. While the LED power is switched every 0.1 s, the

photomultiplier voltage changes only every 5 s because it needs a longer settling time. So a total of 4 sensitivity ranges are alternatively available within 10 seconds. The total sensitivity range is approximately -49...-111 dB.

The backscattering of light from dust has been simulated by computer at CSEM (Centre Suisse d'Electronique et de Microtechnique) within a virtual ice block of 5x5x11 m$^3$ loaded with $2 \times 10^3$ to $1 \times 10^6$ dust particles per cm$^3$ of 2 μm diameter, covering approximately the range of total geometrical cross section of the dust load observed in the EPICA Dome C ice core.

The diameter of 2 μm corresponds to the mode of the measured dust particle size distribution (Potenza et al., 2016). The influence of the real size distribution compared to the assumed single size particles has not been investigated. The simulated average path length of the received scattered light is of the order of 0.5 m, meaning that the model geometry was far sufficient. The expected range of backscattered light for the given detector geometry is -60...-83 dB, which is very well within the specifications of the logger. Possible undesired light propagation between the logger and the borehole wall has

also been simulated and is estimated to be several orders of magnitude below the reflected light from dust in the ice.





**Figure 11: Logger and 3.5 mm Fiberoptic cable winch. A: emitting 400-nm LED. B: Receiving window and mirror.**

## 2.10 Electric power

By avoiding the heavy consumers (breathing air compressor, triplex pump, heat gun) to run in parallel, a 230 VAC single
phase, 4 kW power source is sufficient to run RADIX. At Little Dome C we were in the comfortable situation to use power
from the camp generator.

## 3. Typical workflow for the use of RADIX

Erecting the shelter and installing the workplace for firn drilling is done in one work day. Drilling alone of the firn hole takes
another full work day. However, especially in the first 30 m clogging of the drill by ice chips occurs frequently. The reason



is that at shallow depths the drill may sink too quickly and the vacuum line gets too much ice at once. This requires lifting of the vacuum tube and drill and in the simplest case, the blockage can be removed by tapping it out with a rubber hammer. But often the drill must be heated and dried. So 2 days are typically required for the firn hole. The hole is then logged to check its inclination.

Next, the packer is screwed to the casing using a thread sealant and the pneumatic hose for inflating the packer is attached

and taped to the casing. The packer is inserted into a Plexiglas test pipe with the inner diameter equal to the one of the firn hole. After leak testing of the packer and the threaded connection to the casing with pressurized air, the casing is lowered into the firn hole. Beforehand a tiny amount of silicone grease is applied on the rubber packer, because clean rubber tends to stick strongly on the ice and it can be difficult to loosen the packer at the end of drilling. During lowering, the casing is connected to the vacuum pump in order to remove falling snow from the hole, so that the packer can be placed in an

absolutely clean hole. The packer is set a few meters above the bottom and inflated to about 3 bar above the hydrostatic ice pressure.

Pressurized air is also applied to the casing to check that no porous firn layers are present in the last few meters of the hole. In case of a negative test the hole needs to be deepened. If the test has been favorable the casing is lowered to the bottom of the hole, where the leak test is repeated. If the system is tight, the casing is cut to the correct length, stretched and filled with

fluid. After temperature equilibration the fluid level should stay within a few cm in 24 h.

Installing the fluid system for the RADIX drill and setting up the winch with the hydraulic hose requires another day.

As the body of the drill has a smaller diameter than the casing and as there might be a small gap below the casing, there is some risk that the drill bit could start a hole that is out of alignment with the casing. In order to avoid such offset, a square nut with 20 mm outer diameter is mounted about 5 cm behind the drill bit. The nut guides the drill into a well aligned

starting hole. The nut can be removed after a successful start of drilling.

From now on, the drill could ideally be left in the hole for continuous drilling, preferably in auto-feed mode. The drill would need to be pulled up only for occasional logging of the hole. Even an overnight break without removing the chips from the hole should be no problem. Ice cuttings in the hole seem to move easily after a break. However, at the present state, clogging or anti torque rotation occurred frequently, requiring lifting the drill to the surface for maintenance. Reasons and suggested

solutions are discussed below.

If the drill is pulled up, a small flow through the hose is maintained in order to lift the remaining ice chips to the surface. When lowering the drill back into the hole after maintenance it is important to keep the pump running, because the increasing pressure on the way down might otherwise press some fluid through the drill motor into the hose and force reverse rotation. The fluid in the hole still contains a certain amount of fine ice particles, which might get caught in the motor

and block it. While drilling in a continuous mode, either manually or using auto-feed, the majority of the work arises in connection with the maintenance of the fluid/chips separation system.

A substantial amount of fine chips passes the slotted screen, which is collected in the 50 and 100 μm test sieves. They are changed and cleaned manually at intervals of few minutes. The collected chips as well as the pellets from the screw press are





manually transferred to the heated barrel for melting from where the meltwater is discarded via a bottom faucet. The silicone

fluid floating on top of the melt water is recycled by pumping it back to the barrel passing the test sieves and fine filter.

## 4. Polar field tests 2015 to 2022

The RADIX system has been gradually improved during six campaigns in Greenland and Antarctica at the respective European deep drilling sites.

### 4.1 Renland 2015

A reduced system has been tested on Renland ice cap in east Greenland. The drill hose has been operated manually without a winch. A firn hole from the Danish RECAP project could be used. We attempted to connect the casing at the bottom of the firn hole by freezing it in a small volume of melted ice. A heat cartridge and a camera for monitoring the melting were lowered through the casing. When the lowest 0.1 m of the casing was resting in meltwater the heat cartridge was removed and the water was left to refreeze. The casing was tightly connected to the ice and was filled with fluid. However, drilling

through the end of the casing was difficult because the casing had been deformed by refreezing of the meltwater. Finally the casing was punched by the drill and was no longer tight. A different method to seal the casing was required.

### 4.2 East GRIP 2016

We have used a first version of the hose winch with 500 m of hydraulic hose. We added few brass extension tubes to the drill to increase the weight for better penetration. Melting of the ice at the bottom of the firn hole was done with a dedicated

aluminum end tube with integrated heating elements attached to the PE casing. Melting and setting the casing was quick, but a substantial fluid leak was observed. This leak was caused by the decrease in volume of the ice during cooling, which resulted in a small gap between the aluminum tube and ice. In order to avoid this gap we coated the tube with a 2-mm layer of silicone compound before the melting procedure. A small leak remained after melting and refreezing but immediately disappeared after filling the casing with fluid due to the swelling of the silicone compound by its contact with silicone oil.

We were able to drill about 20 m below the casing.

Despite the additional weight of the extension tubes we still had to slightly push on the hydraulic hose to invoke penetration of the drill. This was probably due to the friction of the twisted hose in the borehole. We suspected that sliding of the anti-torque skates caused twisting of the hose because no rotary union was installed between the drill and the hydraulic hose. The high-pressure Triplex-pump provided too low volume flux because of cavitation at the high altitude of the East GRIP site.

Separation of fluid and chips was done solely with the screw separator, which was quickly overloaded during continuous drilling.



### 4.3 East GRIP 2017

For the first time we have drilled the firn hole with the dedicated mechanical firn drill. The difference to the present version was a smaller, 18 mm/ 15 mm vacuum pipe and we used a small car roots supercharger compressor as vacuum pump.

Especially the smaller diameter vacuum pipe turned out to provide a too small air flow, causing frequent clogging. We changed from a melt connection of the casing to a packer solution. We added the slotted screen vibrator as a first separation step in the fluid circuit. The drill was equipped with a rotary union and the drill bit with a thread-cutting specially ground center bit. We have installed a booster pump before the high pressure pump to avoid cavitation.

For the first time we had a system that seemed to be apt for a productive drilling. Unfortunately drilling was stopped early

due to an internal leak in the packer.

### 4.4 East GRIP 2018

The RADIX system was housed in the Axion tent for the first time as the drilling location was at the margin of the Northeast Greenland Ice Stream, about 30 km east of the East GRIP camp. A 70-m firn hole was drilled with the electromechanical firn

drill connected to a new claw vacuum pump and a 22 mm/19 mm vacuum pipe. The firn drilling was completed in 5 hours. We then encountered problems with PE cuttings from the casing. Finally we succeeded to drill into the ice and for the first time we could drill in auto-feed mode for about 10 m.

At 115 m depth the torque of the motor was not sufficient to drill further. The maximum torque was also partly limited by the relatively high leak flow through the motor. Improvement by a higher flow rate of the high pressure pump was not

possible by the limited volume flux of the booster pump.

First tests with the new logger were carried out. Transmission of the signal was not very stable and the current of the special low temperature batteries was too low. But we obtained first results on the inclination of the hole. Inclination reached up to 20 ° and increased especially where penetration was difficult.

### 4.5 Little Dome C 2019/2020

In preparation for this Antarctic drilling, we mainly worked on improving the torque of the hydraulic motor. We tried several coatings of rotor and stator, but finally found that hardening and nitrating was best suited for our purpose. A substantial improvement was achieved by replacing the previously used Al rod to transmit the rotation from the rotor to the bearing of the drill bit with a flexible shaft and the use of ball bearings instead of slide bearings. We also installed a more powerful booster pump. Various improvements in the fluid/chips separation, e.g. a larger slotted screen, were also implemented.

Firn drilling to 107 m depth and setting the casing was done in 5 days. No refilling of the 7-L compressed air bottle connected to the packer of the casing was needed throughout the whole drilling season.





The start of fluid drilling was difficult because of remains of frozen water in the new hydraulic hose. Normal operation was only possible after heating and flushing the hose with new silicone oil. Most of the drilling was then done in auto-feed mode. However the deeper we got the more the drill needed manual pushing down from the surface. At several occasions the drill got stuck and could only be freed after several hours of pulling with about 500 N on the hydraulic hose. Obviously the static pulling force on the thread cutting center bit was much higher than the dynamic pulling. With increasing depth the motor torque became marginal again. We think this was due to a different cutting regime under high hydrostatic pressures (Yoshino, 2016). At a depth of 246 m we observed some uplift of the hydraulic hose, which we interpreted as a symptom of high inclination and we decided to log the hole.

The unstable performance of the logger observed at East GRIP had beed corrected and a new type of Li batteries with higher current rating was used. Indeed the inclination reached >45° at 240 m depth. The accuracy of the hole azimuth was restricted as the magnetic sensor showed a very high change in offset with temperature.

### 4.6 Little Dome C 2021/2022

With the cancellation of the 2020/2021 Antarctic season due to Covid, a longer preparation period could be used to develop a number of improvements to RADIX to overcome the problems encountered in the first Antarctic drilling. A 50% increase in torque of the hydraulic motor has been achieved by changing the geometry from 3/4 lobes to 5/6 lobes and by redesigning the pressure relieve valve on the drill for a maximum of 70 bar.

In the fluid system we have added plate filters to reduce the amount of small ice particles and added a heated barrel to improve the recycling efficiency of the fluid.

To avoid a rapid increase in hole inclination as in the2019/2020 season we implemented three major improvements: (i) a higher traction of the drill bit resulting from a substantially modified design (Fig. 7), (ii) increasing the weight and decreasing the bending of the drill by replacing brass by tungsten, and (iii) a self-stabilization of the verticality by precisely positioning centering knobs. In an inclined hole, they cause the drill head to tilt against the vertical. The exact position of the knobs has been optimized by numerical simulation and measurement of bending (Fig.12).





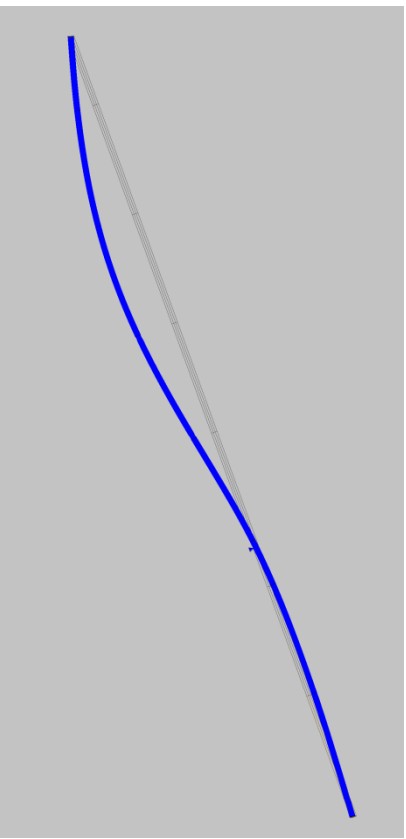


**Figure 12: Simulation of bending of drill by gravity (exaggerated).**

Preparatory lab tests with 0.75 mm pitch thread at -55 °C showed dynamic traction forces of about 180 N per thread turn.
During the drilling at Little Dome C the dynamic traction forces seemed to be somewhat lower, probably due to a less stable
axial control of the drill bit than during the lab tests, which were made with a drill stand.

In the logger we have replaced the magnetic sensor by one with a more than 100 times lower temperature sensitivity of the
offset signal. The azimuth of the logger was very stable and accurate within a couple of degrees.

Overall, the improvements made for the 2021/2022 season have produced the expected results. Drilling from below the
casing at 107 m to the previous depth of 250 m went smoothly in 4 drilling days (Fig. 13).



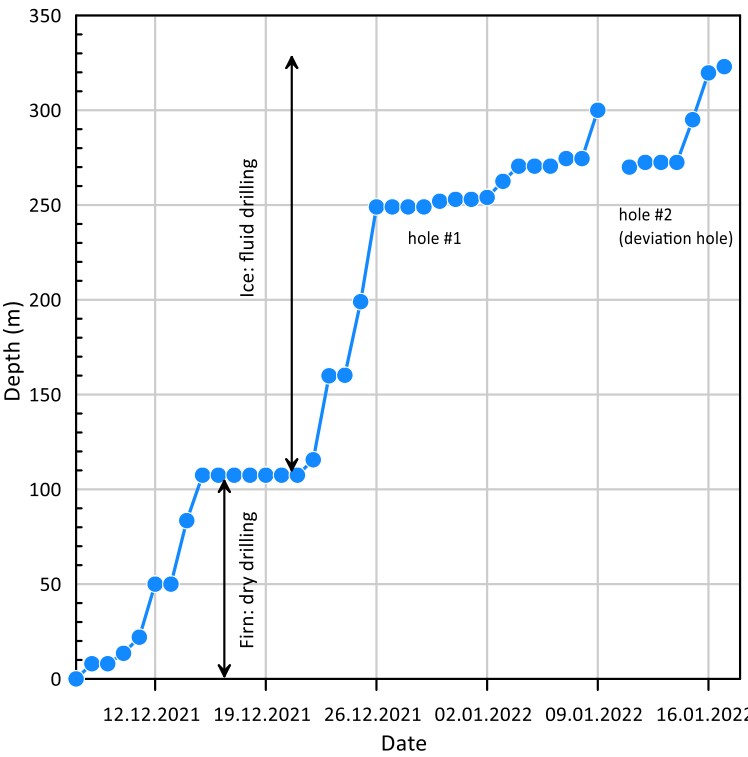

**Figure 13: LDC 21/22 drilling progress**

# 4. Critical appraisal of the latest RADIX deployment 2021/2022

First, the hydraulic hose gave off fine greasy, flaky dirt, which tended to clog valves and the flux controller. After days of
struggling we finally heated and flushed the entire hose for many hours and were able to reduce it to a minor problem.
However, we could not determine the origin of this dirt and why we did not observe it two years earlier, when we had used
the same hose.

The second problem was a too small holding torque of the drill's anti-torque. With the higher torque of the new motor we
obviously exceeded the holding limit of the spring loaded blades. Previously, we had only tested the anti-torque in -30 °C
ice, where we observed ample margin. Additional tests on site showed a rapid decrease in torque in the colder, harder ice.
We then added the spare anti-torque as a second one and improved spring load and edge geometry for maximum torque. The
rotation of the anti-torque repeatedly caused problems in penetration and also deformed the borehole after a few attempts to
initiate drilling at the same depth.

The third problem was the friction of the hydraulic hose in the drill hole. Friction coefficients μ on ice at -50 °C are much
higher than near the melting point, with μ ≈ 0.25 for our Hytrel hose. The drilling fluid has no lubrication effect. On contrary

365



it slightly increases friction compared to the dry case. As the RADIX drill has a very small diameter it bends much easier than a larger sized drill and therefore the curvature radius of the hole is potentially smaller. Especially during phases with difficult penetration the deviations from a straight hole tend to be larger.

Friction amplifies the force on a flexible string that is wound on an object and the amplification grows exponentially, known as Euler-Eytelwein or capstan equation: force amplification = $\exp(\alpha * \mu)$, where $\alpha$ is the wrap angle (Euler, 1769; Eytelwein, 1808). This means that a slight resistance at one end of the hose, through cumulative curvature of the hose, can ultimately completely stop the movement of the hose. This situation was reached at around 300 m depth.

After initially reaching 300 m depth we were not able to lower the drill beyond 280 m. Hole logging showed a sharp turn at 270 m depth and the logger could not be lowered below 278 m due to another sharp bending. We decided to start a deviation hole at 270 m depth. After reaming a straight starting ramp using a countersink with modified, aggressive outer cutting edges, we could successfully initiate the deviation hole with the normal RADIX drill. A 3-D plot of the original and deviation hole are shown in Fig. 14. However, at 324 m depth we encountered the same situation, where penetration was no longer possible due to too high friction in the hole.

Subsequent logging stopped at 316 m where the logger would not further slide down. When pulling the logger up it got caught at this depth, but became free after pulling close to the maximum strength of the fiberoptic cable. This was an indication that the capstan effect could cause severe problems in retrieving a sonde that is even only slightly caught in the hole by reducing the force transmitted to the sonde to nearly zero.

We estimated a total wrap angle of about 90 degrees from 110 to 310 m depth, based on the hole logging data. With a friction coefficient of 0.25, this yields a friction amplification of about 1.5. A 50% increase in friction is substantial for the RADIX drill that is nearly weightless in fluid.



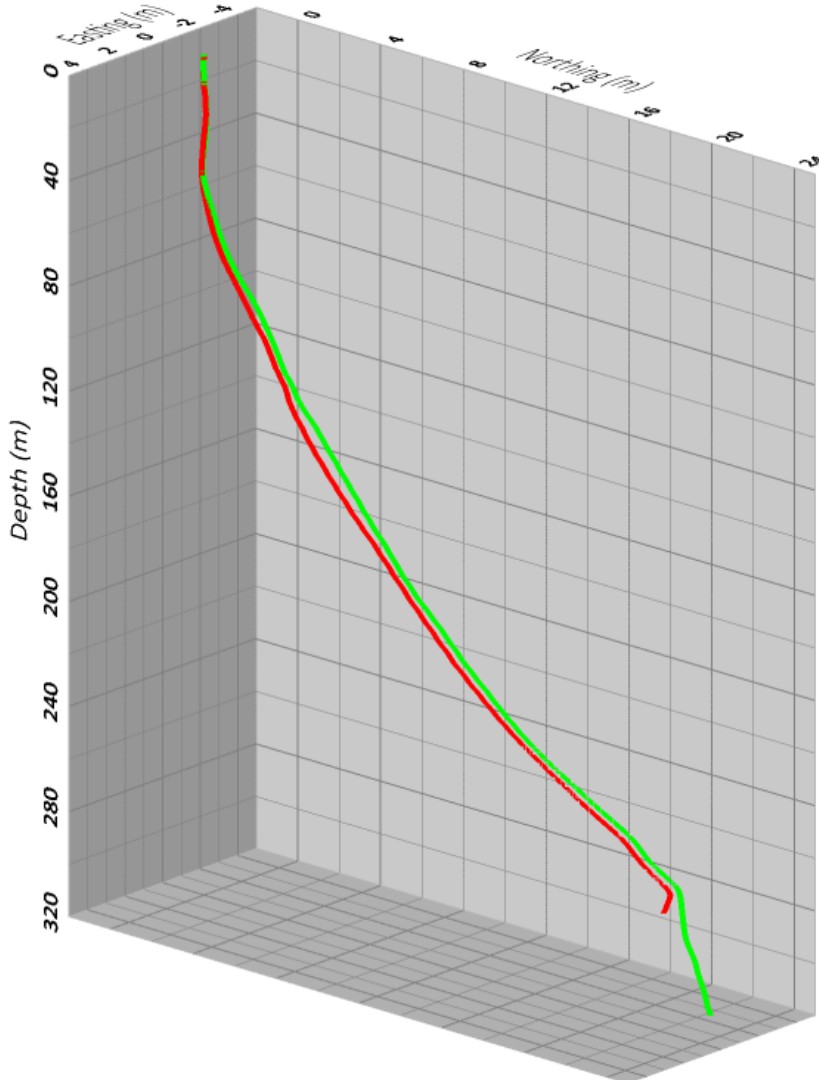

**Figure 14: 3-D orientation of the LDC21/22 boreholes. Red: original hole; green: deviation hole. The difference between the red and green curve, where they cover the same depth, indicates the uncertainty of the determination of the position based on the logger data.**




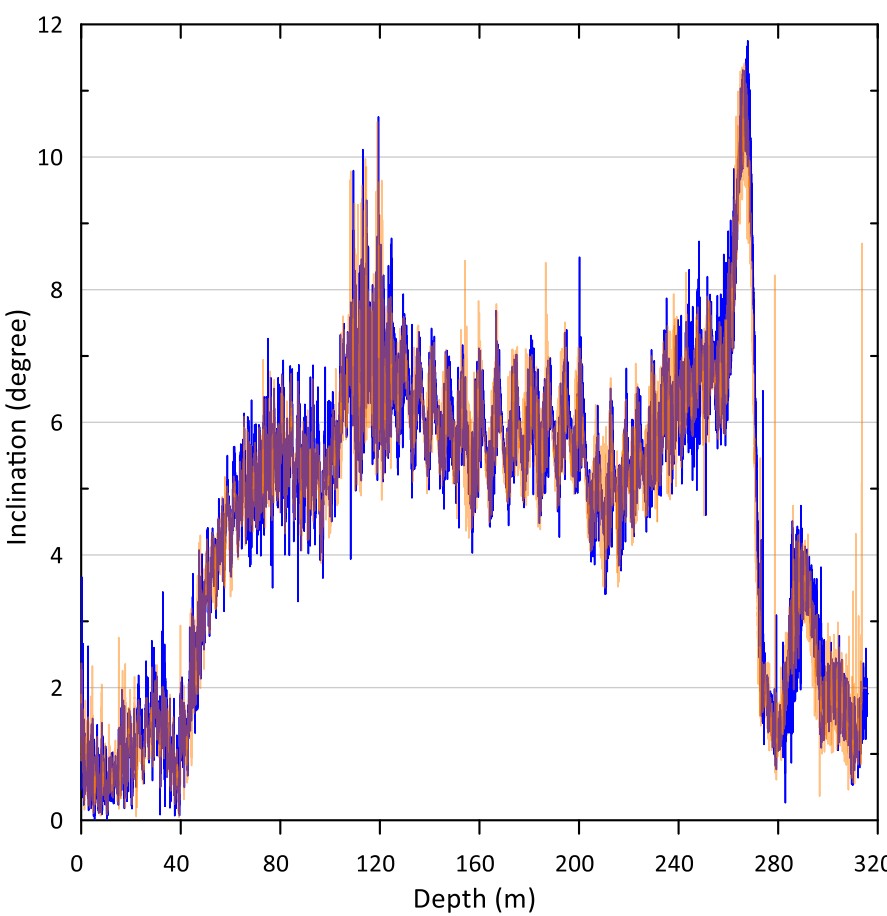

**Figure 15: LDC 21/22 hole inclination (324 m hole). Blue: downward logging; orange: upward logging.**

Fig. 15 shows the borehole inclination including the deviation drilling beyond 270 m. The casing ends with an inclination of
6 to 7 °. We observe a decreasing trend to 5° at a depth of 210 m. The oscillations are a sign of a corkscrewing hole with a
full turn at approximately 6.5 m intervals. Drilling at these depths was smooth and the decrease in inclination could be the
result of the auto correction of verticality by the centering knobs. Below 210 m penetration was slower, which is a sign of
increasing anti-torque rotation. Also, the corkscrewing pitch decreased to about 4 m at 255 m. At that depth we often had
anti-torque rotation and several times we drilled few meters without anti-torque and rotary union (torque taken by hydraulic
hose) in order to reestablish a clean hole for the anti-torque, resulting in a rapid increase in inclination.

Between 266 and 272 m the inclination decreased from 11° to 2°. Rather than an effect of the auto-correction we think that
this decrease is the result of drilling along a big helix that was initiated by the previous penetration problems.

Below that depth the inclination shown in Fig 15 is from the deviation hole. Here we have set the centering knobs 0.3 m
higher up because we suspected too high side force on the drill head resulting in bending the lower part of the drill bit with a
positive feedback on the side force resulting in even stronger bending. By moving the centering knobs to a higher position





we allow more freedom to the lower end of the drill, but losing the auto-correcting of verticality. Still the inclination of the hole remained rather low in the deviation hole and we did not observe oscillations as before.

*Capabilities and limitations of the present configuration:*

- A 40 mm firn drill permits drilling in a semi-continuous mode into the solid ice for placing a casing into a vacuum cleaned hole.
- A lightweight single piece pipe serves as casing and is sealed by a rubber packer inflated with compressed air. The compressed air is prepared on site with a breathing air compressor.
- A fluid system powers the RADIX drill by means of a triplex high pressure piston pump. The fluid is recycled
through various filtering stages.
- 3100 m hydraulic hose on an electric winch provide enough depth capacity for drilling through the entire polar ice sheet at most sites.
- The main drill is driven by a full metal Moineau motor (positive displacement mud motor) with 1.6 Nm torque. It is flow and pressure controlled and housed in a 15 mm pipe.
- A thread cutting drill bit cuts a 20 mm hole at a nominal speed of 10 mm/s.
- Ice cuttings are transported with the drill fluid to the surface and can be sampled as needed for further analysis.
- A deviation hole can be started as necessary at depths with elevated hole curvature.
- A battery powered logger is deployed separately on a fiber-optic cable. It transmits in "real time" temperature, 3-D acceleration, 3-D magnetic field and dust in the surrounding ice. The latter could not yet be fully tested, because it
requires bubble-free ice, which is found only blow the bubble-clathrate transition.
- The main limitation of the RADIX system as used for the Little Dome C drilling 2021/2022 is its restricted depth capability, due to too high friction in the hole.
- Presently, the system needs too frequent maintenance requiring hoisting the drill to the surface. The frequent spooling of the hose additionally lifts and spills too much drill fluid to the surface.

**5. Further developments and Conclusions**

The limiting factor for drilling deeper holes is the too high friction/weight ratio and/or too small curvature radii of the borehole. In order to lower the friction/weight ratio one could make the drill itself heavier by adding more tungsten extension tubes. However a longer drill would be uncomfortable for handling in the present shelter. Another practicable change to the drill would be to increase the diameter, which would however infer a larger hole diameter and a whole rat-tail of changes, i.e.
abandoning the concept of minimum weight and minimum resources. Another possibility to increase the weight would be to use a different material for the hydraulic hose. In the present configuration the total weight of the hose immersed in fluid with a density of 930 kg/m³ is approximately 700 N. An armor of steel instead of aramid or even a plain steel tube would add

sufficient weight for moving the drill down. However, a heavier hose or plain tube would entail major changes to the winching device with the consequence of stronger, bulkier equipment.

Concerning lowering the friction in the hole, there are two main sources of friction, the hose and the anti-torque. As stated above, in order to maintain the low-weight concept, we must stay with a plastic hose. We have found that the friction coefficients of common plastics on ice at -50°C are rather similar and since friction amplification increases exponentially with the wrap angle, a change of a few percent in the friction coefficient would not make much difference.

At Little Dome C, we had to use a double anti-torque with substantial spring loads on the skates to get sufficient torque. The
axial friction of the anti-torque made up a substantial part of the weight of the drill and still the torque was marginally sufficient for drilling. For a future version of RADIX the anti-torque system must provide higher torque while reducing the axial friction.

A reduction of the total wrap angle is probably where the greatest potential for improvement lies. As flexibility of an object does not scale linearly with its dimension, small objects tend to flex excessively, so does RADIX, potentially producing
curved holes. A larger diameter drill would help, but this would, as mentioned, entail changes toward heavier equipment and require more drilling fluid. In order to reduce the wrap angle with the present dimensions we must introduce specific measures. These include geometric considerations in the design of drill and bit, and reducing the flexibility of the lower part of the drill in order to avoid helical paths of the drillhole.

*Author contributions*. JS developed and designed the overall concept of RADIX. RW and SM improved design details and realized mechanical components. JS, TS, and RM participated in field campaigns in Greenland and Antarctica. JS wrote the manuscript with contributions from all co-authors.

*Competing interests*. The authors declare that they have no conflict of interest.

*Acknowledgements*. We would like to thank Hanspeter Moret for implementing the controller software. We are grateful to Jörg Pierer, Centre Suisse d'Electronique et de Microtechnologie, for providing an extensive simulation of the light scattering on dust particles in ice. The logger electronics was realized with great care by Spacetek (www.spacetek.ch). We gratefully acknowledge the long-term support by the Swiss National Science Foundation (SNSF).

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
