# Peer review of "Progress of the RADIX fast access drilling system"

_The Cryosphere, 2022_

## Author Comment (AC2)

**Authors responses to Reviewer1 Comments on tc-2022-183**

Reviewer 1: John Goodge
* * *
"Progress of the RADIX fast access drilling system" by Jakob Schwander et al., The Cryosphere Discuss.
* * *
**GENERAL COMMENTS**

**RC1: I'm pleased to see this report on development of the RADIX drilling system and the details provided by the authors. Success of this system has been widely anticipated, and the manuscript gives a thorough overview of the technical development and field results. This will be of interest to many researchers in Greenland and Antarctica. This manuscript describes both the technical parameters of the drilling system, results of successive field tests, and looks to the future in terms of ways to improve the system. The authors provide a balanced look at the design aspirations with the field deployment hurdles, giving an honest and transparent appraisal of the system.**

**The abstract is concise and clearly written, giving a clear picture of how the system operates. If space allows, I suggest inserting a second sentence with a statement of the drill's purpose or goals; this would come between the opening statement that a drill has been constructed and the following description of what it does. It would be good to include the name of the drill in the abstract, following that in the title. For example, "The goal of RADIX is to provide rapid, deep access for sampling of ice and down-hole physical measurements that inform about ice sheet age and history."**

Reply: We have added the suggested statement and the name of the drill in the abstract.

**RC1: Although the abstract gives an appropriately broad overview, after identifying the design depth it does not say that field testing in Greenland and Antarctica achieved only about 300 m. I think it's important that this contribution lay out the system design, the system operation and modes, and finally the results of the field tests, as the authors have done. Yet both the goals and the testing outcome should be stated in the abstract so there is a clear distinction between them.**

Reply: We have stated in the abstract "Routine performance has been established to a depth of 300 m" As this depth has only been reached in the last field test, we have changed the statement to "The greatest depth reached so far was 324 m during the last field project at Little Dome C"

**RC1: I think the manuscript is very solid and deserves publication. My strongest recommendation is to streamline the discussion of the design and operation to make it more accessible to a wider audience, as noted in my comments below.**

**SPECIFIC COMMENTS**

The authors provide a very thorough review of the technical elements of the RADIX system, and the manuscript is packed with information. My overarching concern is that it seems the target audience for this paper is not defined. Is it the relatively small number of people who also work on developing ice-drilling technologies, or is it a wider audience of people who use this type of technology (engineers, drillers and scientists) and yet want to know the basis for how it works even without having to sift through every detail? The first group will be quite small whereas the latter group is considerably larger and will give the paper a wider reading. As discussed below, there are in my view too many details and not enough focus on the broader scope of the technology.

Some of the descriptive information is perhaps overly detailed (e.g., how the packer is set up and deployed). In many places the text reads like an operations manual, which could be very helpful if there were several versions of RADIX in use. However, I'm not sure this level of detail is necessary in such an article to convey the important intrinsic parameters or properties of the drilling system so that someone can still understand the process by which the drill works, without getting bogged down in how every tube is connected. The detail is quite dense and makes for slow reading. Some of this technical presentation could be streamlined to focus on the most critical aspects of the system, and the remaining details put in an appendix or an operation manual made available in an online archive.

Reply: We have tried to present the topics in a more balanced way. Since RADIX is not yet an established system and the recent field experiences are essential for the presentation of the current state of development, we have refrained from separating these experiences into an appendix or supplement.

RC1: An example of this over-detail is description of how the hydraulic motor was made (starting line 142). The production of this motor and cutting head is magnificently done, but is it necessary for most readers? If someone is interested in taking on such a project themselves, can they not contact the authors directly or look for this information in an appendix?

Reply: The motor and its production as well as the cutting head are the heart of RADIX. We have now emphasized this in the text. They deserve therefore a bit more space.

RC1: Many of the figures (including photos and CAD drawings) would benefit from labels. Only a couple of the figures with photos of components are labeled. For example, I can't see the anti-torque skates in the photo of Fig. 5. Presumably they are right below the black hose, but the photo isn't well enough resolved to be sure. Where are the flow controller and hydraulic motor in this photo? Likewise, please label described components in Fig. 6 and other figures.

Reply: We apologize that the figures in the submitted manuscript are at relatively low resolution. We have improved several figures and added labels.

RC1: A discussion of how the hose behaves against the borehole wall is only discussed toward the very last part of the manuscript. I think this issue could be brought up earlier in the discussion of field tests, then revisit during discussion of future improvements. It is certainly relevant to this and similar drilling approaches whether there are any frictional issues as the previously coiled hose makes contact with the borehole. In addition, is there ever a problem with hose kinking or any other behavior during unwinding?

Reply: The friction between hose and borehole emerged only as a problem at the very end of our field tests. As stated above the recent field experiences are essential for the

presentation of the present state of development. At the moment, the list of possible difficulties is not yet complete and we cannot yet rank their importance.

**RC1: TECHNICAL COMMENTS BY LINE NUMBER**

**I provide many specific comments and suggestions below, intended to help with clarity and wording usage, nearly all of which can be fairly easily addressed by the authors.**

**8. Use of forward is OK, given explanation, but in the drilling industry this is also referred to as 'normal' circulation, as opposed to reverse circulation where fluid is returned up the drill pipe.**

We have changed to "normal", so that it matches common usage.

**9. Perhaps make it clear that the logger is deployed into the borehole after the tubing is removed?**

Done.

**11. It would be good to define 'quickly'. Can you give a penetration rate, either as a target or a range?**

Done.

**16. Need to define WAIS.**

Done.

**17. Deep ice-coring projects are particularly expensive and are in contrast to the drilling technologies like RADIX that this paper is about. It's important to distinguish coring projects — whose entire aim is to extract intact cores — from more rapid and less expensive drilling methods, many or most of which support reconnaissance for coring targets. So here and in next few lines, important to distinguish these two and replace 'drill' with 'core' in some of the references.**

Changed to make difference clear.

**21. Suggest adding '… minimal overall resources and weight during field deployments.'**

Added

**75. General recommendation about writing. Although many of the experiences being reported happened in the past, I strongly recommend writing sentences like this in present tense. Doing so makes the writing more active and also more generally conveys how the system works, not just how it worked. So here, "… the tent provides a calm interior … One door is usually left open…". There are some cases when it's better to use past tense (e.g., "We collected ice chips from hole #2…"), which refers to specific cuttings in a specific hole. Otherwise, present tense is better for telling the story of how things work.**

Thank you for this recommendation. We have switched to present tense where appropriate.

**88. Glazes, not glaces.**

Corrected.

**95. Is C-VLR a common term? Need to define?**

It is the pump type. We put it in parenthesis.

**109. Why was a 3 bar overpressure chosen? Is this always sufficient to maintain a seal?**

In theory, the seal is tight in a smooth hole when the overpressure is higher than the pressure needed to inflate the rubber hose to the hole diameter, which is below 1 bar. With 3 bar we have a good safety margin for pressure variations.

**139. What is 'leak flow'?**

Definition and details added.

**146. After a number of details about manufacturing, this statement about the motor rotation is unclear. It seems to pertain more to operation. What is the purpose of the eccentric movement? Does that aid in fluid transport, or rotation of the drill?**

The eccentric movement lies in the principle of the Moineau motor. We have separated and explained in an own short paragraph.

**155. Because the RADIX design is based on a coiled tube format that pays out over 3000 m, the properties and behavior of the hydraulic hose during cold working conditions seem important but overlooked here. How flexible is the hose at low temperatures? Does is risk kinking? Have any problems been encountered?**

Due to its small diameter the hose stays flexible. There is no tendency for kinks.

**160. What type of depth counter? Electronic? Manual? This is an important feature during operation, so surprisingly lacking in explanation.**

Changed from " a depth counter" to "an electronic encoder"

**161. Change wording from "than the one of the drilling fluid" to "than that of the drilling fluid". Change speed to velocity.**

Done

**161-164. This discussion of feed rate is confusing. How does sinking velocity relate to coiling rate? How do these related to feed rate? What is the feed rate in typical operation? Several facts and quantities are provided but it's not clear how they all relate to the control of feed rate, which is presumably a measure of penetration rate.**

We have improved the wording in this section

**166. Suggest completing the thought here… "We use … oil as the drilling fluid."**

Thanks, done.

**172. Think should be "wedge wire screen"? Invert words. Same should be changed in caption to Fig. 9.**

Yes!

**190. Because a borehole packer is placed after drilling through firn in order to create a seal, this is perhaps a confusing re-use of the term packer here. Would it make more sense to refer to this as a 'gasket' or 'ring'?**

Wording changed

**198. Can authors provide more detail here about where the sampling occurs? Is this done by cutting material off of the screens? That seems to be the first opportunity prior to going into the press. How are materials sampled from "specific depths"? That is, is it a matter of knowing flow rates combined with drill depth to calculate the sampled depth? Further explanation would be helpful.**

This is a important point. Since we have not yet drilled to deeper and interesting levels, we don't want to extent this discussion very far. We have added a few statements.

**200. Several questions come to mind regarding the borehole logger. Above it is noted that the borehole is evacuated of fluid by swiping upward with a packer, so presumably (but unstated) is that the logger is designed to work in an evacuated hole. The borehole is said to be 20 mm and the OD of the logger housing is 15 mm, so there is a 2.5 mm annular gap between the logger and borehole wall. It would be good to remind the reader of these dimensional comparisons here. This is important because it's helpful to know how the system is expected to work if the logger is not in direct contact with the borehole wall and there is an air gap between. In that case, are there any issues with signal transmission by the laser optical dust logger (is there signal loss or scatter)? And if the sonde and borehole wall are not in direct contact, how does that affect the measured T profile? Is the borehole allowed to equilibrate thermally before being logged? If so, how much time is allowed? What is the rate of descent and is the T signal dependent on this rate? In essence, does the logging rate allow for continuous T measurement even when the sonde is not in**
**contact with ice? In general, some additional details of explanation about the logging process would be worthwhile. [Note, reading further, method for evacuating the hole is described before method of logging, but it's not clear if in practice the borehole is evacuated before logging or if logging is done via fluid-filled hole. This sequence should be clarified.]**

We have changed the order of subchapters 2.7 (now "ice sampling") and 2.8 (now "fluid recovery"), but we leave 2.9 ("Bore hole logger") behind the hole drilling chapters because it is a separate unit, but we now state that logging is done in the fluid filled hole. We had mentioned that undesired light has been model simulated. We have added that there is a 2.5 mm gap between logger and borehole wall.
As the principal component of the logger, the dust sensor, could not be fully tested we plan to publish more details in a separate paper, after a full test.

**205. Electronics "are" housed…**

Corrected.

**208. Should be "notebook computer".**

Added.

**238. Drilling of firn depends, of course, on thickness of the firn. This can vary quite a bit in Antarctica and be well over 100 m. The firn drill is specified to 120 m, so presumably this isn't a problem completing in the period described, but here again some specifics would be helpful (e.g., "We can complete 120 m in XX hrs.")**

We have changed to wording to be more specific.

**256. Here and elsewhere, should distinguish between a day (24 hr) and a work day. If a step in the workflow can be completed in a work day (8-12 hrs) that is very different from a full day.**

We have changed to "work day" where appropriate.

**262. But 'occasional logging' would require the borehole to be evacuated?**

Logging is done in fluid filled hole.

**263. Perhaps you mean 'stoppage' rather than 'break'?**

Corrected.

**303. Best to delete "have". Because this is not a report of most recent activities, best to express in simple past tense "we drilled…". Including 'have' is an example of using passive voice and should be avoided. This applies elsewhere as well (search and destroy the word 'have').**

Deleted here, but we don't agree that perfect tense is passive voice. Nevertheless, we have searched the paper for "have" and deleted where inappropriate.

**304. No idea what a "car roots supercharger compressor" is. Common terminology?**

We have simplified to "roots-type supercharger" s. e.g. on Wikipedia

**310. Statement leaves the reader hanging in suspense. Was this a defect in the packer, or a loose hose connection? One is an easy fix, but if there is a defect in the packer it would be good to know the root cause.**

We said that it was in the packer, not a hose connection. We have added that there was an air leak.

**323. Important details not given here. The intended design is to make a vertical borehole. If the hole ended up inclined from vertical at 20°, what is the cause of this deviation and how can it be prevented in future. Treat this as a learning experience and let the reader know why this occurred.**

We added a sentence, telling what we thought was the reason for the increasing inclination.

**340. Misspelling of 'been'. Later, batteries 'were' used. And again, what do you make of the high angle of inclination here?**

Corrected. We added a statement regarding the high inclination.

**344. Future readers may not get this reference. Better to be explicit and say "COVID-19 pandemic".**

Corrected

**356. Neither in caption of Fig. 12 nor in text are the parameters of the model defined. How was this simulation constructed?**

We have added some details in the caption to Fig. 12. We think the rest is self explaining

**366. Figure 13 needs a more expanded caption to describe what is shown. Rather than use full numerical dates (which have various formats internationally), simply label as Dec 12, Dec 19, etc., and state in caption or label figure that this period spans 2021-22. Might be appropriate to invert the depth axis.**

We switched the date to the ISO 8601 format YYYY-MM-DD. Depth orientation of the depth scale is a matter of taste (look at it from the Northern Hemisphere☺)

**368. It would be appropriate to open this section with a general statement that after prior drilling tests a number of issues were encountered that have either been addressed or remain as problems for future consideration. It's helpful to have a topical statement of what is being said. Based on the summary later, suggest making this section 4.1 Problems encountered (section 4 to remain as "Critical appraisal...").**

We prefer to keep the chapter as it is since the information is based on the last RADIX deployment in Antarctica

**384. Change "known as" to "following the".**

Adopted.

**388. From these field reports, it seems that the maximum practical limit to drilling is ca. 300 m. Is this correct? Even if not, it would be helpful to the reader to comment here on why the depth limit does not correspond to the design limit (3000 m). What is the source of this? Can it be overcome?**

This is discussed in the Chapter 5

**402. In Figure 14, caption states that over a similar depth interval there is some uncertainty about position (which might also include orientation?). Are the original and replicate holes actually so close in orientation (given by easting and northing directions)? It seems unlikely that the replicate hole would so closely follow the original?**

The replicate hole only starts at 270 m depth by deviation from the original hole.

**410. So the variation in inclination can vary by as much as 2° (shown by oscillations) just due to corkscrewing of the hose? If true, does this mean that the borehole is non-linear at a scale of a few meters?**

Noise on the submeter scale is due to shaking of the logger when moved. Statement added in the caption to Fig. 15.

**424. This is a good summary of the field findings. Make this section 4.2 so that it stands out more prominently.**

We prefer to keep the chapter as it is a concise bullet list as part of the "Critical Appraisal" and the status of our knowledge after the last deployment in Antarctica.

**440. "Blow" should be "below".**

Corrected

**443. Suggest deleting "too".**

Done

**446. Again, here delete both uses of "too". That is a qualitative judgement and sufficient to say here that author's view that a limitation comes from high friction and small curvature.**

Done

**449. "Rat-tail" is perhaps too colloquial a term. Change to 'cascade'?**

Adopted.

**451. Weight (mass) is not measured in newtons, but kilograms. I think the meaning here is that by increasing the weight of the drilling tube the resulting force pulling the drill downward and applying force on bit would be greater, but this needs to be corrected.**

Scientific definition of weight: "the weight of an object is the force acting on the object due to gravity" and therefore Newton needs to be used here.

---

## Author Comment (AC3)

**Authors responses to Reviewer2 comments on tc-2022-183**

Reviewer 2: Julius Rix
* * *
"Progress of the RADIX fast access drilling system" by Jakob Schwander et al., The Cryosphere Discuss.
* * *
General Comments:

**RC2: Great to see an excellent paper reporting on the continuing development of the RADIX drilling system. This gives a good update from the original Annals of Glaciology RADIX paper and compiles the various seasons in Greenland and Antarctica where the system was tested, the problems encountered, and the modifications carried out to try to overcome problems.**

**Very impressed with the heated plumb bob and the firn drill, both of which may have many other applications in the future.**

**Likewise, the fluid recovery technique is unusual and could be used in other drilling systems.**

**The borehole logger itself is very interesting and in my opinion merits a technical paper of its own. Both the dust logging and the orientation measurements are of interest. It would have been nice to see the dust logger working in the bubble free ice.**

**For an ice core drilling engineer my feeling is that the amount of detail is at about the**
**right level for a wide audience. Being a driller there were times when I wanted more detail but realise that not all of this journals readers will be quite as interested. Having said that, it might be more appropriate for the field reports (section 4) to be placed into a supplement. Although the field reports are essential to understand the development of the system, this is probably only of interest to the drilling community.**

Reply: Since RADIX is not yet an established system and the recent field experiences are essential for the presentation of the current state of development, we have refrained from separating these experiences into an appendix or supplement. We feel that the chronological reporting provides the reader with a better picture about the current status, or "work in progress", as also suggested by the title of the paper.

**RC2: All in all a very interesting paper for a very ambitious drilling system. The conclusions seem to point to a larger diameter (less ambitious) drill being a solution. Hopefully this knowledge can be usefully used in the future.**

**Specific Comments:**

**Line 32 Colis – should be packages?**

Changed to packages

**Line 43 m3/h – superscript on the 3**

Done.

**Line 50 5/6 – maybe should be 5 or 6, could be read as 5/6ths. Having read further it would be good to explain what you mean by 5/6 lobes.**

5/6 lobes is common terminology for this type of motor, but we agree that it may be difficult to interpret for the major part of readers. We have added an explanation on line 138.

**Line 55 m3 – superscript on the 3**

Done.

**Line 106 3/2mm – should be 3mm/2mm to avoid confusion with 1.5mm**

Done

**Line 110 liter – should be litre**

Corrected.

**Figure 6 I'm a bit confused about how the flow controller works, I'm assuming that the flow is from the top downwards, but how does the flow pass the spring-loaded pin?**

There are vertical holes in the body around the pin

**Line 32 15/5 mm – should be 15mm/5mm to avoid confusion**

Done.

**Line 138 5/6 – could be 5 or 6 to avoid confusion**

Explanation added

**Line 160 by a depth counter on the sheave – is this an encoder?**

Changed to encoder.

**Lines 165-186 and figures 8 and 10 – The figures have consistent wording and numbers which is great however the description does not. For instance, the 150 L drum (clean reservoir) is called the Fluid container with level switch in figure 8. It would be good to make these consistent with each other.**

Thanks. Made consistent.

**Line 338 some uplift – is this the hose coming off the sheave wheel suggesting that the hose is jamming in the hole?**

We agree, this is unclear. Wording improved.